# Student Approaches and Performance in Element Sequencing Tasks Using 2D and Augmented Reality Formats

Kieren H. McCord [1,*], Steven K. Ayer [1], Logan A. Perry [2], Karan R. Patil [1], Jeremi S. London [3], Vanessa Khoury [4] and Wei Wu [5]

[1] School of Sustainable Engineering and the Built Environment, Arizona State University, Tempe, AZ 85283, USA; sayer@asu.edu (S.K.A.); krpatil@asu.edu (K.R.P.)
[2] Department of Civil and Environmental Engineering, University of Nebraska-Lincoln, Lincoln, NE 68588, USA; logan.perry@unl.edu
[3] Department of Engineering Education, Virginia Polytechnic Institute and State University, Blacksburg, VA 24061, USA; jslondon@vt.edu
[4] School of Engineering, Lebanese American University, Byblos P.O. Box 36, Lebanon; vanessa.khoury02@lau.edu
[5] Lyles College of Engineering, California State University Fresno, Fresno, CA 93740, USA; weiwu@csufresno.edu
* Correspondence: kieren.mccord@asu.edu

**Abstract:** In civil and construction engineering education research, a focus has been on using 3D models to support students' design comprehension. Despite this trend, the predominant mode of design communication in the industry relies on 2D plans and specifications, which typically supersede other modes of communication. Rather than focusing on the presentation of less common 3D content as an input to support students' understanding of a design, this paper explores more common 2D inputs, but compares different visualization formats of student output in two educational interventions. In the first intervention, students document a construction sequence for wood-framed elements in a 2D worksheet format. In the second, students work with the same wood-framed design, but document their sequence through an augmented reality (AR) format where their physical interactions move full-scale virtual elements as if they were physically constructing the wood frame. Student approaches and performance were analyzed using qualitative attribute coding of video, audio, and written documentation of the student experience. Overall, results showed that the 2D worksheet format was simple to implement and was not mentally demanding to complete, but often corresponded with a lack of critical checks and a lack of mistake recognition from the students. The AR approach challenged students more in terms of cognitive load and completion rates but showed the potential for facilitating mistake recognition and self-remediation through visualization. These results suggest that when students are tasked with conceptualizing construction sequences from 2D documentation, the cognitive challenges associated with documenting a sequence in AR may support their recognition of their own mistakes in ways that may not be effectively supported through 2D documentation as an output for documenting and planning a construction sequence. The results presented in this paper provide insights on student tendencies, behaviors, and perceptions related to defining construction sequences from 2D documentation in order for educators to make informed decisions regarding the use of similar learning activities to prepare their students for understanding the 2D design documents used in industry.

**Keywords:** augmented reality; education; hands-on; sequencing; construction management

## 1. Introduction

Accurately understanding a building design is critical for making effective decisions. The architecture, engineering, and construction (AEC) domains require teams of professionals to collaborate to effectively design and build infrastructure. This collaboration has

traditionally involved the communication of complex three-dimensional (3D) concepts using two-dimensional (2D) plans (i.e., "blueprints"). This 2D mode of communication has been used for hundreds of years and continues to be used as the format for contractual deliverables to enable stakeholder communication [1–3]. To be prepared for this kind of interaction, it is important for students to understand how to define construction processes based on their design comprehension.

Recently, educational researchers have focused on how 3D modes of communication can lead to effective design comprehension by construction students. Some have explored the use of emerging visualization tools such as augmented reality (AR) as resources for performing various construction-related tasks [4]. Augmented reality is "any case in which an otherwise real environment is "augmented" by means of virtual (computer graphic) objects" [5]. Some have even compared student performance between 2D and AR conditions, where users relied on one modality or the other as a design resource for performing construction-related tasks [6]. One group compared 2D and AR as modalities for presenting a design, and with a written questionnaire, found much higher design comprehension when receiving the design in an AR modality [7]. Others have explored the advantages of enhancing 2D documentation with AR content for field-based tasks with industry practitioners [8]. These studies illustrate some of the ways that AR may offer benefits compared with 2D documentation to support design comprehension, but they focus on the use of AR as an input or resource to support this understanding. In this paper, an activity 'input' is any resource provided to inform a student's understanding of a design concept, and an 'output' is the resultant product that students deliver upon completion of the activity. In this case, the output constitutes the student's defined means of creating and documenting their actual construction sequence.

This study focuses on student behaviors and perceptions when given 2D plans as a design input or resource used to define a sequence for each piece of wood required to construct a section of a wood-framed wall. In this study, each intervention explores a different mode of visualization output, with both still using the 2D input, that students may use to see the results of their process: either a 2D worksheet (Intervention 1); or an AR-based model (Intervention 2). While 2D materials are common classroom educational tools, AR is an emerging tool that is much less commonly used and has not been the subject of extensive study in a classroom context. This work identifies ways in which students' construction sequencing processes and perceived experiences compare when they view their defined processes in these differing formats.

Research shows that design comprehension can be developed through experience. A study by Hölscher & Dalton [9] revealed that people with actual experience in architecture were much better able to understand design complexity in 2D documentation (as opposed to video presentation). Another study concluded that design comprehension a skill that can be developed through experience, not just ability [10]. In construction research, Dadi et al. [11] found that workers in construction fields who frequently leveraged drawings in their work were able to use drawings to perform assembly tasks more efficiently and confidently than their less experienced counterparts. In addition to technical skills development, hands-on, authentically situated experience in the construction fields has been known to provide motivation to students as well as to helping develop independent problem-solving skills [12]. Given these clear advantages of authentically situated learning experiences, providing students with access to these activities would develop the skills and attributes that would best prepare them for design comprehension and success in industry. From these works, it can be concluded that students who have authentic learning experiences, or experience in building certain types of construction, can generally do this effectively. While most educators understand these advantages, the realities of higher education classrooms are such that most students do not have access to authentic, hands-on learning activities in the classroom. This limitation is likely due to the high cost and resource requirements needed to bring these types of activities to a class or lab [13]. Given the known advantages of authentic learning but the extremely limited access, this work seeks to understand the

ways in which students perform element sequencing tasks when using AR to simulate the physical interactions of authentic learning with the low cost of traditional design communication, comparing with traditional 2D outputs as a baseline for what may be expected in a traditional classroom.

This work provides context to the broader learning community of educators who are considering leveraging various modalities to teach relevant construction skills. For the outputs for the student work, the 2D worksheet was chosen to replicate a typical classroom situation, and AR was chosen to explore an emerging technological tool with the potential to bring authentic, active learning into the classroom, as it mimics the kind of environment and physical movements that may happen on an actual jobsite. Presenting the task virtually through AR and not physically with real materials was chosen to explore the viability of AR as a low-cost, reusable alternative to using and discarding real construction materials. The results presented here provide insights to allow more informed decision making when considering AR and 2D methods for teaching construction sequencing tasks based on 2D documentation.

## 2. Literature Review

### 2.1. BIM and 2D in Construction

Building information modeling (BIM) and other advanced 3D visualization tools have become increasingly common in the construction industry [14,15]. Advanced immersive visualization techniques, such as virtual reality (VR) or AR, have been proposed to complement 3D modeling [16] and help in the visualization process, such as for assembly tasks [17,18] or for enhancing spatial cognition in construction practitioners [19,20]. Much of the literature discussing visualization of construction documentation explores the possibilities of advanced technology such as mixed reality, with the assumption that these more interactive models offer superior features, such as flexibility and communication clarity [21]. Overall, interest in 3D advanced visualization tools is seeing a steady increase in the AEC domains.

In addition to exploring 3D and more immersive modalities, some researchers have looked at ways to leverage traditional 2D drawings, exploring ways to facilitate the process of drawing interpretation, such as using AR as a supplement to traditional drawings [8,22]. Other researchers have explored the conversion of 2D drawings into 3D models with varying levels of automation [23–28]. These works focus on the relationship between 2D and 3D documentation, but not necessarily on the impacts to the users or their learning.

In addition to industry-based research, extensive research has gone into exploring the integration of BIM in higher education [29]. Other research has aimed to identify the best approach to incorporating BIM into the construction engineering curriculum, comparing 3D to 4D and 5D [30]. These advanced, and 3D, modes of communication have been shown to support learning gains such as by increasing participation, interaction, and motivation [31], and by teaching specific skills and competencies [29,32].

### 2.2. AR in Engineering Education

AR is an emerging visualization tool that has been explored in a variety of modalities to support learning. For example, educators can use mobile devices like smartphones or tablets [33], or head-mounted displays [34] for presenting AR content. Most educational research has focused on smartphone and tablet-based AR, likely due to the accessibility of these devices. In fact, two recent literature reviews on AR in construction related disciplines and AR in engineering education disciplines both reported that only 5% of educational studies explored head-mounted pass-through AR [35,36]. In addition to the choice of user technology, there are also various levels of interactivity reported in literature regarding AR for engineering education. A recent review of augmented reality applications in engineering education revealed that very few published research papers on AR applications (2%) provided users the opportunity to interact with the application by using gestures [36]. Most favored device movement, controls within the application, pen pointers, or marker

positioning. In the engineering education review paper, only two articles of the 42 studied allowed users to assemble virtual elements, both within the electrical engineering discipline [36]. These works illustrate the opportunity for researchers to advance the body of knowledge by understanding the ways in which head-mounted AR and gesture-based AR interactions support engineering education contexts.

Looking more specifically within the construction-specific discipline, many studies have explored the use of emerging visualization technologies like AR as resources for helping students learn or demonstrate skills, with some researching AR alone, and others comparing AR performance with 2D resources. For example, researchers have studied how AR as an input can affect spatial reasoning [6], cognitive load [37], deviation detection [4], and drawing interpretation [8]. These studies explored AR as an input to support users' design comprehension rather than an output, or a means of design creation for those users. In one study, student users were tasked with physically assembling electrical and plumbing elements while using either a head-mounted AR display or 2D drawings as inputs that showed them where to physically place the elements [38]. These researchers found that students preferred the AR input and were able to more efficiently complete the assembly task. While these works made significant contributions to the understanding of AR as an input for design comprehension, there is a need for studies that explore AR as an educational output, where AR does not replace 2D documentation as a resource, but rather explores AR as a classroom tool to simulate hands-on learning before students enter the physical workforce.

Other construction-adjacent fields, such as architectural design, have begun producing research that indicates value in exploring AR and 2D as educational outputs. In this field, researchers have compared virtual reality and paper-based design as educational design tools, finding that the virtual environment led to higher enjoyability and likelihood of use [39]. In another study comparing AR and 2D, researchers explored differences between students completing a 2D design activity and an activity based in AR and found that enjoyment and interest levels were similar between the two formats, but the 2D activity led to design fixation [40]. These prior works demonstrate how AR outputs can impact students' learning processes in design contexts, but a similar study is missing from the literature for construction contexts.

In summary, the prior works presented in this literature review illustrate the range of studies that have leveraged AR as an educational tool in engineering education in general as well as specifically in AEC domains. The work presented in this paper is novel in its use of full-scale, head mounted AR, enabling students to physically interact with virtual materials in a hands-free manner, providing closer simulation of physical construction activities than handheld alternatives. The level of interactivity of the AR application is novel, as it departs from the majority of studies that use AR as a visualization tool with only static, graphical elements or with limited user controls. The technological choices made in this paper were chosen to most closely approximate hands-on learning to explore AR as a simulation of a learning strategy with known benefits. Many prior studies, especially in AEC disciplines, explore the use of 3D modes of communication as inputs to provide to students to support their design comprehension in order to complete a learning task, but far fewer studies explore how 3D augmented reality modes of communication as outputs may affect students' learning. This work departs from these prior studies by utilizing AR as the means by which students experience the process of visualizing their construction sequence outputs, based on their understanding of traditional, 2D documentation for design information. This study's use of AR as a full-scale, interactive learning output is first-of-kind in the construction discipline and is compared to a 2D output to provide a baseline for typical classroom activity outputs. Prior studies that do explore how 3D augmented reality learning outputs impact students' behavior focus on learning related to design, not construction. Given the need for construction professionals to be able to define effective construction sequences, and also the prevalence of 2D documentation in the construction industry, a better understanding of how 2D and 3D outputs affect

student learning based on traditional 2D inputs would enable educators to purposefully use these tools to prepare their students for success. The identified literature gaps inform the research questions that frame the study in this paper and are as follows. When utilizing 2D documentation as an input to perform element sequencing activities:

- RQ1: What behaviors and perceptions do students exhibit that indicate learning in a 2D output format?
- RQ2: What behaviors and perceptions do students exhibit that indicate learning in an AR output format?
- RQ3: What comparisons regarding potential impact on student learning can be drawn between using 2D and AR learning formats for defining construction sequences?

## 3. Materials and Methods

### 3.1. Overview

This paper presents a study where students utilize 2D design documentation to determine the element sequencing of wood studs within a wall section. In this exploratory, uncontrolled experiment, two groups of students participated in an intervention, with the first group of students documenting their results on a 2D worksheet, and the other group in AR. The task was selected to resemble a type of carpentry work that is common on jobsites. The scope of the design was limited to a single section of wall to enable participants to understand and perform the task within a single session and within one hour. An elevation view from the 2D documentation provided to students is shown in Figure 1, and the other drawings are included in the Appendix A.

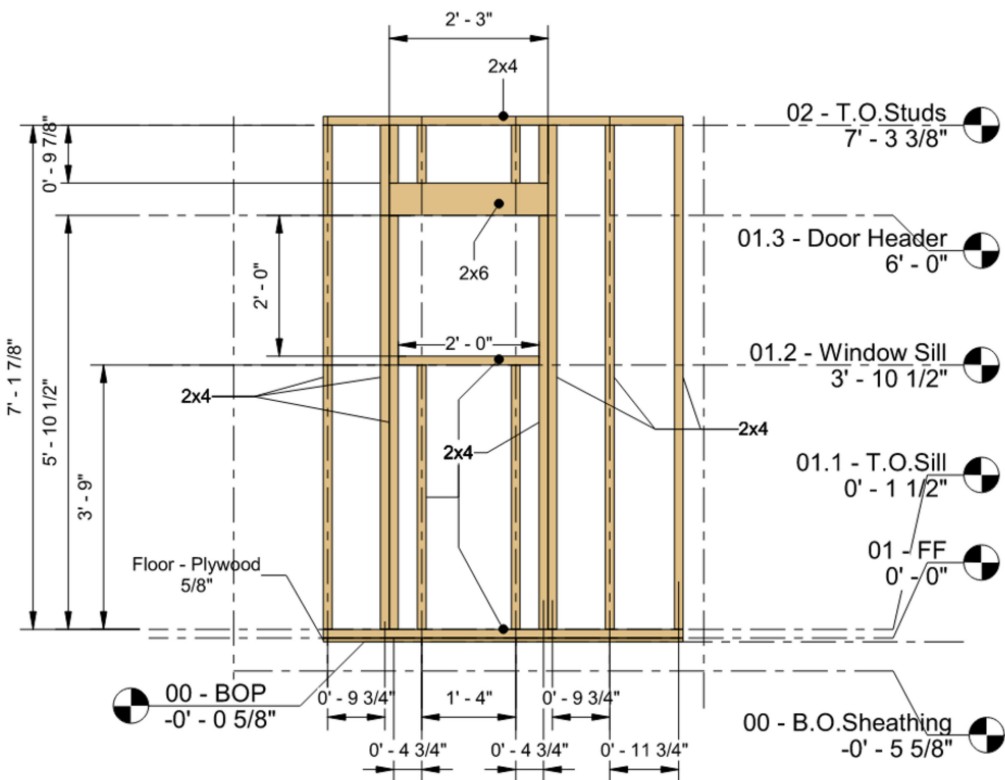

**Figure 1.** Example of one of the drawing sheets provided to students: a window elevation view of the framed wall. See the Appendix A for the full set of drawings.

Both interventions studied began with a pre-activity survey that targeted students' prior experience and demographic information. Then, students were tasked with interpreting the 2D documents provided in order to define a construction sequence in either a worksheet or AR. In the study presented in this paper, head-mounted AR was chosen to give students an AR experience that did not require holding a phone or tablet, freeing

up their hands for interaction with the virtual construction materials. In this way, the application presented in this article affords a unique and enhanced level of interactivity that departs from previous work in this field, where students use gestures and voice commands to interact with full-scale elements, all of which are fully interactive. After completing the sequencing activity, students were asked to complete a post-activity survey to understand their perceptions of their performance. The processes involved in this work are displayed in Figure 2.

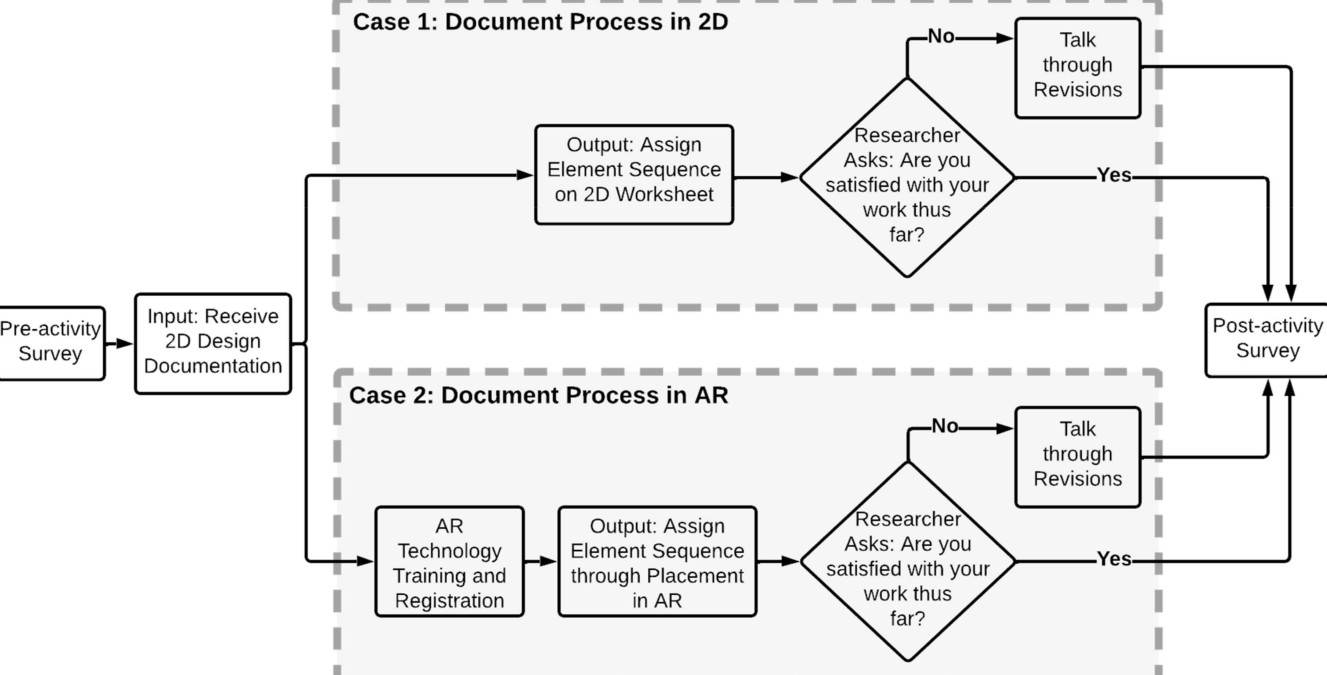

**Figure 2.** Flowchart outlining the steps involved in the two learning activities.

Participation in these learning exercises was offered as extra credit for a construction course and took place over two years, with Intervention 1 (2D Output) implemented first and Intervention 2 (AR Output) implemented the following year. Both interventions were impacted by the COVID-19 pandemic, which necessitated some form of videoconferencing for supporting interactions between student participants and researchers. Intervention 1 involved students defining their construction sequences on a 2D worksheet via a videoconferencing and screen-sharing application. Intervention 2 involved students defining their construction sequences while using the head-mounted AR device alone in a room on campus with a researcher interacting via videoconferencing, in accordance with health and safety restrictions.

### 3.2. Pre-Activity Survey and Reception of Design Documentation

Before completing the activity, students in either intervention took a survey that collected demographic information and asked them to evaluate their skill level regarding various wood-framing related tasks on a 5-point Likert scale. The skills students were asked to self-evaluate included:

- Understanding design and construction documents;
- Finding information (sizes and dimensions) on design and construction documents;
- Deciding on means and methods for installing a structure based on the documents;
- Defining a sequence for installing wood framing components;
- Installing wood framing components correctly.

After the pre-activity survey, students were given the set of drawings shown in Figure 1 and in the Appendix A, and then they were instructed on what to do for each of the activities.

### 3.3. Intervention 1: Construction Sequence Documented via 2D Worksheet

Students were given a worksheet and told that the drawings corresponded to the design on the worksheet, which contained a list of pre-cut lengths of wood and an isometric view of the built wall, with empty spaces for assigning the pre-cut pieces (Figure 3). For the students to know the lengths of the pieces on the worksheet, they needed to cross reference the design drawings. They were asked to verbalize their sequence, think aloud about their process, and annotate the page if they wanted to keep track of any relevant decisions or notes. After completing the sequence, they were asked if they were satisfied with their work and were given the opportunity to make any changes.

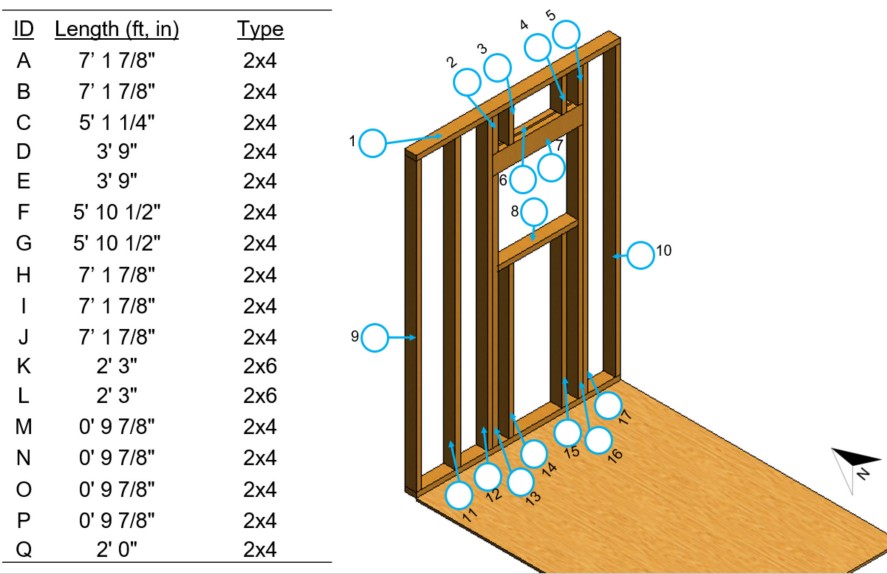

| ID | Length (ft, in) | Type |
|----|-----------------|------|
| A | 7' 1 7/8" | 2x4 |
| B | 7' 1 7/8" | 2x4 |
| C | 5' 1 1/4" | 2x4 |
| D | 3' 9" | 2x4 |
| E | 3' 9" | 2x4 |
| F | 5' 10 1/2" | 2x4 |
| G | 5' 10 1/2" | 2x4 |
| H | 7' 1 7/8" | 2x4 |
| I | 7' 1 7/8" | 2x4 |
| J | 7' 1 7/8" | 2x4 |
| K | 2' 3" | 2x6 |
| L | 2' 3" | 2x6 |
| M | 0' 9 7/8" | 2x4 |
| N | 0' 9 7/8" | 2x4 |
| O | 0' 9 7/8" | 2x4 |
| P | 0' 9 7/8" | 2x4 |
| Q | 2' 0" | 2x4 |

**Figure 3.** The worksheet where students assigned items on a list of pre-cut wood studs (**left**) to the numbered spaces on an isometric view of the completed wall (**right**).

### 3.4. Intervention 2: Construction Sequence Documented via AR

For the AR experience, an application was developed utilizing both physical and virtual resources, as shown in Figure 4. All wood components were presented to the students virtually through AR via a Microsoft HoloLens 1. The wall model was designed in Revit and the components were imported into Unity. Scripts within Unity were designed to enable users to move pieces with hand gestures and install them with voice commands.

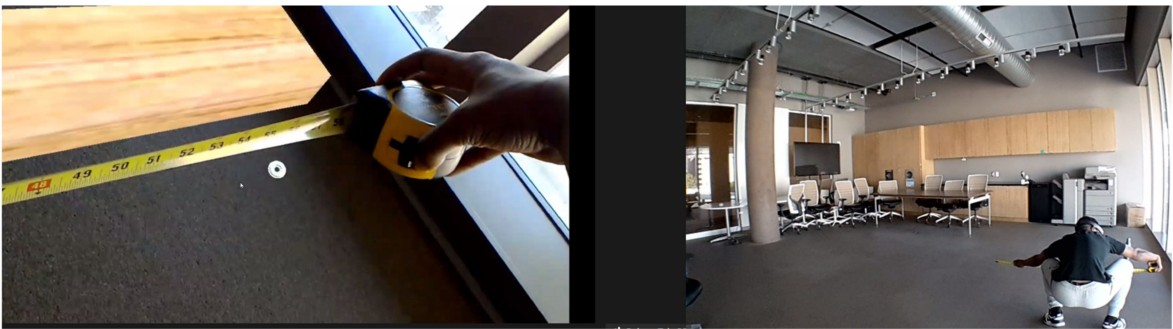

**Figure 4.** (**left**) A piece of virtual wood being measured with a physical measuring tape and (**right**) a third person view of the student measuring the virtual wood (the wood is not visible unless wearing the AR device).

Students were instructed on how to wear and use the AR device by the researchers who joined via videoconferencing. Students were guided through the process of opening the application, registering AR content in the room, and interacting with virtual building materials. More specifically, the researchers instructed the students on how to measure the virtual floor plate using a physical measuring tape and to verify that it matched the design dimensions, then move it to its appropriate location with a hand gesture, and install it with a voice command. This training enabled students to successfully learn and demonstrate each technological interaction that would be required during the AR construction process. As they participated in the training, students were aware of the upcoming task and how these actions would prepare them for their anticipated task.

After the AR training was completed, students were tasked with defining a construction sequence for the virtual wall elements to correspond with the drawings (see Figure 5). After the placement of the initial piece during training, the researchers allowed students to complete the activity on their own and did not provide any guidance to the students on how to read or otherwise interpret the 2D documents provided to them for completing the construction sequencing task. To complete the task, students relied on the provided set of 2D drawings, a measuring tape, and a pen to mark the drawings. Additionally, students could access a virtual array of wood laid on the ground. The virtual building materials were not placed in any particular order, similar to how materials are often found on real construction sites. As students completed the construction sequencing task, they were asked to think out loud during their process. Each student had one hour to complete the task. When only 10 min remained, the researcher stopped the students and asked them if they felt that the work they completed was correct thus far. At this point, students were also asked if they believed that their progress thus far was done correctly and they were given the opportunity to explain any changes they might make if they had more time to build.

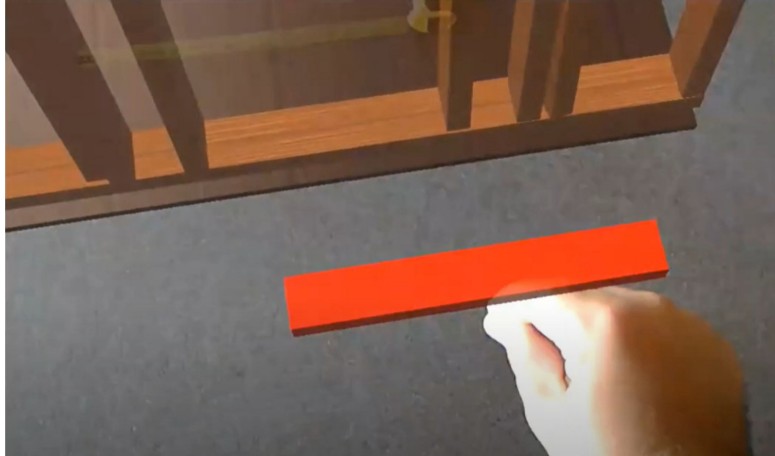

**Figure 5.** A student moving a piece of virtual wood toward a partially assembled wall.

### 3.5. Post-Activity Survey

After students indicated satisfaction with their construction sequence in either of the interventions, all students were provided with a post-activity survey to reflect on the process. They were asked about the cognitive load of the activity using a selection of questions derived from the NASA-TLX questionnaire [41], including their mental demand, their level of frustration, and their perceived success in the activity. In the second part of the survey, students were also asked open-ended questions for each activity. For the worksheet activity, they were shown a video demonstrating the use of the AR activity tested in Intervention 2 and asked to think about the worksheet activity they did, then about the AR activity they saw, and to reflect on how they think their experience and process might differ if they performed the same activity in the other environment. For the AR activity,

they were asked an open-ended reflection question regarding what they learned from the virtual building experience.

### 3.6. Analysis

The pre- and post-survey responses were saved and the students' behaviors and performance were observed in situ and video and audio were recorded for later analysis in accordance with the Institutional Review Board (IRB) approval. The survey results, open-ended question answers, and video recordings served as the basis for analysis. Following models of qualitative coding from similar studies in this domain, a triangulation of participant think-aloud statements, survey responses, and researcher observations provided an understanding of the overall experience. Within the AEC domains, studies that have followed similar approaches in comparing two groups using a combination of qualitative and quantitative metrics including the group who compared AR and 2D inputs for construction assembly tasks [38], and researchers who compared VR and AR for design review [42]. In this study, the videos were coded using qualitative descriptive and attribute coding with the method outlined by Miles, Huberman, and Saldaña [43]. During this process, a series of codes or descriptors are developed that help in extracting meaning from participants' words, actions, and characteristics. Multiple researchers—three for the 2D output and two for the AR output—worked together on creating and validating the codes to ensure consistency in the process. As an example of the codes that emerged, in the 2D output format, two emergent codes were 'length mistake' and 'header mistake', referring to when students placed a wood piece of the wrong length, or placed a piece incorrectly for the window header, respectively. After the codes were created, they were considered together to see what patterns or meaning could be extracted from the experiences. For example, the 'length mistake' and 'header mistake' were considered together in the thematic category of visualization difficulties. The major patterns that emerged through this process are described in the following section along with resulting descriptive statistics. Some basic statistical analyses are added to complement the qualitative analysis described above for the pre-activity questions regarding wood framing experience and the post-activity questions from the NASA-TLX survey regarding frustration, mental demand, and perceived success. An independent samples t-test was conducted for each of these parameters between the two groups, where the unit of analysis was the student response. Given the preliminary nature of these analyses and this study, we chose a smaller sample size that would focus on the qualitative aspects that emerged while still sufficiently meeting the qualification of normal distribution for an independent samples t-test. The statistical explorations were undertaken to augment the qualitative exploration that forms the basis of the paper.

### 4. Results

### 4.1. Descriptive Statistics

The 2D activity involved 38 undergraduate student participants with varying levels of experience in both school and industry. The dataset included 1 freshman, 5 sophomores, 15 juniors, and 17 seniors. Self-reported racial and ethnic backgrounds included the following: Black or African American = one, Hispanic or Latino = 11, White = 23, Other = two, Prefer Not to Answer = one. Finally, 31 participants self-identified as male and seven as female. Regarding wood framing abilities, the average self-rated scores for participants on 1–5 Likert scales were 3.8 for 'understand design and construction documents', 3.9 for 'find information such as sizes and dimensions on design and construction documents', 3.0 for 'decide on means and methods for installing a structure based on the documents', 3.3 for 'define a sequence for installing wood framing components', and 3.2 for 'install wood framing components correctly'.

For the AR activity, a total of 15 undergraduate student participants successfully participated in the protocol, including two sophomores, 10 juniors, and three seniors. The racial and ethnic responses to the survey included American Indian or Alaska Native = one, Asian = two, Black or African American = two, Hispanic or Latino = three, White = six,

Prefer Not to Answer = one. The self-identified genders of the students included 12 males and threes female. Regarding wood framing abilities, the average self-rated scores for participants on 1–5 Likert scales were 3.7 for 'understand design and construction documents', 3.9 for 'find information such as sizes and dimensions on design and construction documents', 3.0 for 'decide on means and methods for installing a structure based on the documents', 3.2 for 'define a sequence for installing wood framing components', and 3.1 for 'install wood framing components correctly'. Notably, these averages were very similar to the average abilities reported by the group involved in Intervention 1 (2D Output), with no statistical difference in any of the categories in independent sample t-tests, demonstrating comparability between the two samples regarding self-reported abilities on wood-framing-related tasks.

### 4.2. Emergent Themes

The participant think-aloud statements and reflections are combined with researcher observations from the videos to identify components of the activity that may affect the potential for classroom implementation and learning and are presented thematically. From this analysis, the major themes of interest are presented. For the 2D Output, the two major emergent themes that are discussed in this paper include Visualization Difficulties and Extemporaneous Approach (see Table 1). For the AR Output, the two major thematic categories that emerged include Visualization Enablement and Self-Regulated Approach (Table 1).

**Table 1.** Themes and codes observed within student approaches and performance for the 2D and AR Output Experiences.

| | Theme | Supporting Evidence (Codes) |
|---|---|---|
| 2D Output | Visualization Difficulties | Leaving Mistakes Unfixed |
| | | Verbally Stating Visualization Difficulty |
| | Extemporaneous Approach | Time Spent Per Piece |
| | | Choosing Not to Revise |
| | | Asking No Clarification Questions |
| AR Output | Visualization Enablement | Physically Fixing Mistakes |
| | | Verbally Recognizing Mistakes |
| | Self-regulated Approach | Time Spent Per Piece |
| | | Referencing Resources |
| | | Verbally Stating Learning Outcomes |
| | | Asking Clarification Questions |

Supporting evidence and clarification regarding the themes and codes presented in Table 1 are shown subsequently, organized by theme.

#### 4.2.1. Theme: Visualization Difficulties (2D Output)

In the 2D worksheet format, a common theme that emerged was visualization difficulties, where students struggled to maintain an accurate mental model of the process they were planning. One prominent indicator of visualization difficulties was mistakes made and left unfixed. A mistake in this context refers to when a student chose a piece from the schedule of materials and indicated their intention to place it in a specifically labeled location where the item length did not match the dimension indicated by the drawings. It was observed that over half of the students (53%) made at least one mistake when determining their sequence.

Examples of commonly occurring mistakes include when students chose the wrong length for a vertical stud, when they selected the wrong dimension for the top plate, and

when they selected the wrong type of piece for the double header, or did not realize there was a double header (see the Appendix A for the drawings). For some of these mistake types (the top plate and the double header), the students would have to refer to the plan views in addition to the elevation view to obtain a complete understanding of the elements. The dimension of the top plate is indicated on all the plan views, but not on the elevation view. The size of stud for the double headers is 2″ × 6″, where all other studs are 2″ × 4″. This designation and the back-to-back placement of the double headers are only shown on the "Floor Plan-Header" view of the wall design. The visual representation of the double headers is partially occluded in the worksheet isometric view and fully occluded on the first sheet of drawings. While all drawing sheets were provided to students as a set, it appears that there was some difficulty in utilizing all drawings together, evidenced by the abundance of errors on the worksheet. Difficulty accessing information that can only be found by accessing multiple drawing sheets has been reported as a potential issue even for industry practitioners [8], and this issue seems to be fully present in student learners when using a 2D visualization format. While some students eventually corrected their mistakes during the process or in their final review, they still noted visualization difficulties in the process. Furthermore, more than half of the students who made mistakes (29% overall) finished the exercise with at least one error remaining unfixed and unrecognized upon completion.

In the activity reflection, where students discussed how their process might differ if they performed the AR task, visualization difficulties were a common theme. In fact, 29% of students mentioned visualization challenges in their reflections. Many suggested that keeping track of pieces and the installation order was difficult. For example, one student said "on paper it was hard to remember what I had put in place and what was still in the lay down" providing evidence that keeping track of an updated mental model of the defined construction sequence was a challenge for students. Many students also recognized their own propensity for making mistakes in the 2D worksheet format, suggesting that in AR, "I would be able to see the errors I was making instead of guessing if I was on the right path" and that "I would physically be able to install each piece and see where it goes. I would also be able to visualize where I made a mistake and why I made that mistake. This would be able to provide immediate feedback". One student summarized their perceptions on the struggles of the 2D worksheet and the potential for AR using their personal industry experience as a factor, stating "With an augmented reality experience the activity would be much less conceptual. Without the experience I have in architecture and construction it would have been difficult to correctly identify which pieces go where in the activity as well as what order to install them in. The augmented reality allows you to have a hands-on experience and visualize the construction process which is incredibly useful when learning about construction". These comments and ideas principally served to illuminate the difficulties students had in forming and updating mental models, but also provide ideas of what themes and evidence may emerge in the AR intervention.

4.2.2. Theme: Extemporaneous Approach (2D Output)

Most students who completed this exercise did so quickly, and typically without second-guessing or making critical checks. Notably, all students completed the activity well within the allotted timeframe. The average time students spent actively completing the activity was 9 min 8 s, with a range from 2 min 47 s to 21 min 40 s. Considering the average time and the fact that all students placed all pieces, the average time spent per piece was 32 s. The high speed of completion provided extra time for students to reflect, identify errors, or fix mistakes, but despite this time, no students revised their final sequence when provided the opportunity. This may indicate either low ownership in the task, high confidence in the correctness of their selections, or the inability to identify their mistakes. Based on post-activity survey data, this decision was likely tied to high confidence, with most students perceiving their work as highly successful, regardless of errors (see Section 4.3). For example, one student—who still had mistakes in their

process—when given the opportunity to review and revise their work, either did not notice or decided not to change the errors and stated "I think I'm good". In addition to the speed and lack of revision, the researchers noted that most students did not reach out during the process to ask questions, choosing to complete the task independently, even though the facilitator was available at all times. Even among industry practitioners, failure to make critical checks and to pay attention to details [44] has been reported as a challenge with traditional documentation and 2D formats, and these results provide evidence that a 2D output does not remedy these challenges. Based on Thorndike's 'trial and error' theory of learning, learners are motivated to continue performing actions that lead to success and avoid actions that lead to failure [45], suggesting, as many researchers have since, that there is value in encountering and learning from failures [46]. This is only possible when learners understand when they are succeeding or failing, and this 2D worksheet approach did not provide evidence to suggest the self-awareness that leads to valuable learning.

### 4.2.3. Theme: Visualization Enablement (AR Output)

From the observational data, it was noted that the AR process seemed to facilitate student visualization in order for them to recognize and fix their mistakes. For example, one participant placed a stud on the wrong side of one they had already placed. When the participant reviewed the piece compared to the drawings, the individual immediately recognized and fixed the issue, stating "I accidentally installed the wrong piece . . . so I had to uninstall and move it. I should have measured first, but I didn't". This comment illustrates a lesson learned by this student, recognizing an error in their process that could potentially inform future work. Other students made similar length errors and corrected them, either by gazing at pieces they had already placed that showed a mismatch in length or by comparing their built model with the paper plans (Figure 6). In addition to recognizing mistakes made during the process, some also recognized mistakes at the end of their building time while evaluating everything they had built. For example, one participant, when asked if they were satisfied, said that they noticed a stud they had placed was not the correct length. The vast majority of students in Intervention 2 (AR Output) recognized their mistakes at some point in the process, with only one student (7%) leaving unrecognized length errors. Two students (13%) left the exercise with one or two pieces rotated incorrectly, which should be noted but cannot be compared to the mistakes made in the worksheet (Intervention 1), since orientation was unchangeable in the 2D format.

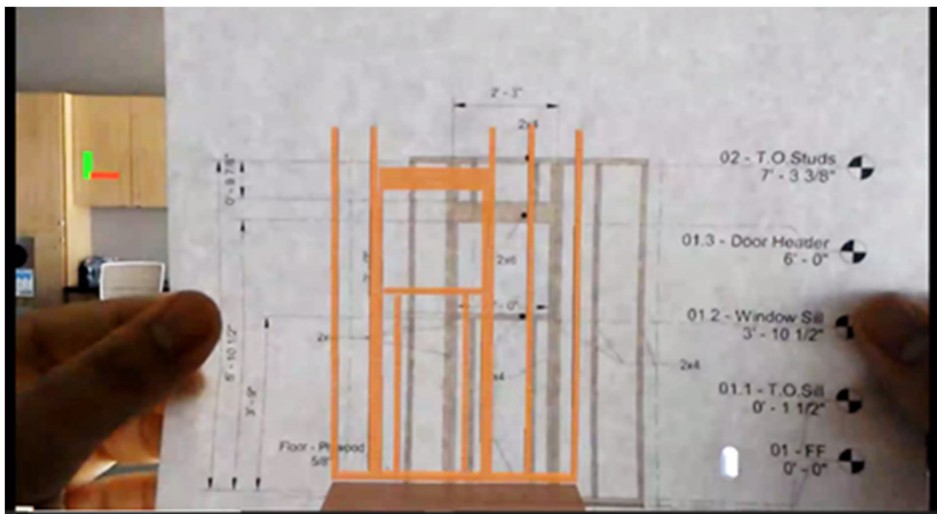

**Figure 6.** A student compares the paper elevation view drawing with their partially constructed virtual wall.

In addition to inferential evidence of visualization enabling mistake recognition, students also made comments regarding how the activity explicitly enabled learning

through visualization. One student suggested that this activity may have direct real-life applicability, stating "I learned what it looks like to frame a wall … I learned that the process is a lot easier to understand with all of the moving pieces in front of me. Seeing the moving parts with the plans all together made the whole thing make more sense. I can actually picture in my head what those plans would make in real life, and have a better picture of how I would go about developing those plans". Others made comments suggesting the potential for this modality to help teach lessons with an impact outside the classroom, such as the student who remarked that they "learned to expect challenges" and another who said that their confidence "really did improve" through the process. These findings align with John Dewey's 'learn by doing' theory, which states "we learn only because after the act is performed we note results which we had not noted before" [47], meaning that when students see and understand their performance, how they achieved those results is solidified in their learning.

### 4.2.4. Theme: Self-Regulated Approach (AR Output)

The students took a much slower, more methodical, approach to completing the activity in an AR format, likely facilitated by the nature of the interface. In this activity, most of the students did not finish placing all pieces of the wall. Some got close, with one student placing all but the top plate and the plywood backing, while a few spent their time organizing and moving the pieces, but never actually installed any. Notably, the time needed for the setup and registration process varied between students, which determined the amount of time available for active material placement. To objectively understand the rate at which students actively built, the average amount of time spent placing each piece during active building was calculated, resulting in an average of 3 min 26 s taken per piece placed. This average is notably higher than the 32 s per piece from Intervention 1 (2D Output). It is important to note that while both interventions similarly required the students to verify dimensions of existing pieces and to assign a construction order to these pieces, the mechanism for doing so is different between the two. Verbalizing and/or writing a series of numbers is different from physically having to move each one into place. Therefore, the comparison of time per piece is not intended as a measure of which task is superior or inferior, but rather to provide a descriptive understanding of the difference between the two experiences.

In addition to the speed of the activity, it was noted that many students exhibited a pattern of referencing their resources. For example, many took the time to check the drawings multiple times throughout their process. Many also measured each piece before placing it, with some even taking the time to verify the measurements of pieces after installation. Even students who did not get close to finishing had the potential to study and utilize the 2D documentation, with some mentioning learning gains like a student who stated that the activity "helped me understand a lot about building plans that wasn't clear before. I think I learned more about reading drawings with this experiment [than] even the physical building process". They mentioned how physically moving the pieces into place while looking at the drawings helped them understand the drawings better, even though they only had time to place a few pieces.

In addition to leveraging physical resources, many students took the opportunity to ask questions of the facilitator, some of which related to the process of framing a wall, with students sometimes asking the facilitator if their work was correct thus far or asking for help reading dimensions on the drawings. While the researcher did not intervene to answer specific questions that would positively or negatively impact the students' processes during the activity, their interest in asking these types of questions suggest engagement with the task.

### 4.3. Post-Activity Survey Results

The thematic analysis illustrated differences in students' abilities to visualize and detect mistakes as well as distinct approaches to each task. In addition to the thematic

analysis, the post-activity survey answers to the cognitive load (based on NASA TLX questions and scaling detailed in the methods section) and performance-related questions give further insight into these differences, including mental demand, frustration, and perceived success. Figure 7 presents these results for the two interventions graphically, side-by-side.

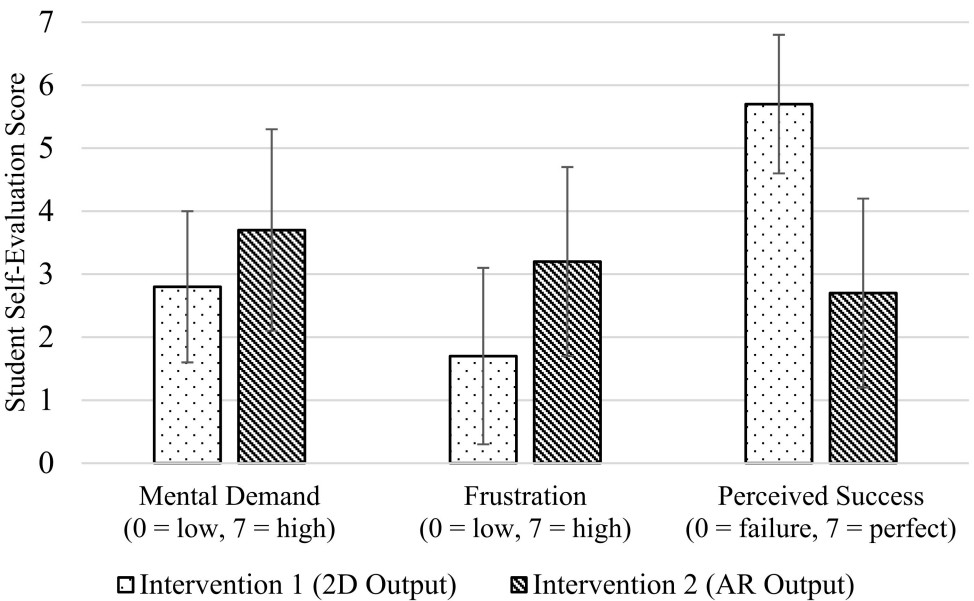

**Figure 7.** Post-activity survey results of students' self-evaluations regarding the mental demand (0 = low demand, 7 = high demand), frustration (0 = low frustration, 7 = highest frustration), and perceived success (0 = failure, 7 = perfect) of the 2D Worksheet activity (*n* = 38) and the AR Experience (*n* = 15). Averages are shown by the solid bar values and standard deviations are presented with the error bars.

For Intervention 1 (2D Output) students rated their mental demand at 2.8 on average (0–7 scale), and their frustration at an average of 1.7 (0–7 scale), which are both well below the midpoint of the scale. These results suggest that students perceived the task to be easy and not frustrating to complete, despite mixed actual success. For the AR activity, mental demand was rated at an average of 3.7—just above the midpoint—and frustration at an average of 3.2—just below the midpoint. In both measures, the ratings were higher for AR, suggesting that this activity requires more effort and engagement from students in terms of thinking and overcoming frustrating challenges. Some of the frustration could be due to using and learning a new technology, while some might be attributed to the challenges of the sequencing task itself, and this study does not claim to control for these differences, only noting that the overall AR experience presented higher frustration to students than the 2D worksheet counterpart. For both measures, the differences between the two interventions were significant ($p < 0.05$ for mental demand, $p < 0.01$ for frustration) using an independent samples t-test. All variables met the assumption of normality needed to run this statistical analysis (skew < 1 for all variables involved).

Regarding students' perceived success on the activities, the average rating for the worksheet activity was 5.7 on a 0–7 scale. Interestingly, this score was very similar for students who finished the activity with an unresolved mistake (5.6) and for those who finished the activity with no apparent mistakes (5.7), which offers further evidence to indicate that this method may not be ideal for students to engage in trial and error learning, since recognizing mistakes is critical to that process. On the other hand, students in the AR activity rated their success at an average of 2.7 on a 0–7 scale, indicating much lower satisfaction with their performance. It is hypothesized that both an awareness of mistakes and not finishing the activity may both have contributed to the low success rating. The

difference in perceived success between the two interventions is statistically significant ($p < 0.001$) using an independent samples *t*-test.

## 5. Discussion

The results of the thematic analysis benefit from contextualization in the learning theory literature. In particular, the behaviors observed during the AR activity point to the potential for learning based on both Thorndike's 'trial and error' theory of learning and Dewey's 'learn by doing' theory. Although considered opposing theorists on some aspects of educational reform [48], elements of both Thorndike and Dewey's learning theories persist in some forms today and motivate work in experiential learning. For either approach, the potential to affect student learning is predicated on students' abilities to identify success and errors by understanding output, which leads to a refinement of their behavioral inputs. In other words, it is important for students to have an opportunity to act for themselves, have successes or make mistakes, and recognize those successes or mistakes. The results from this study provided evidence that these critical checks are not easily facilitated through the 2D input-2D output experience, but are readily facilitated by the 2D input-AR output student experience.

Regarding the survey results, perceived success approximates confidence in a task, and confidence in graduating students is an asset, but perhaps just as important is knowing where knowledge falls short. Often, educators only look at success scaffolding, but failure scaffolding—intentionally exposing students to failure in the learning process—is an emerging research area that may be just as critical to truly internalizing lessons and using that knowledge in meaningful ways in the future [49]. Using an AR output to help students conceptualize construction sequencing processes from 2D plans may support this process. These activities utilized minimal instructor involvement after initial instruction, a situation often used by design to promote independent learning [50,51]. When working independently, students encounter both successes and errors and ideally learn from both. As foundational educational theorist Dewey stated, "by doing" an activity, the student "becomes familiar with its methods" and "acquires needed skill" [47]. The AR activity emerged as the learning environment that facilitated the performance awareness that allowed students to demonstrate high potential for learning by doing. The survey results align with the approaches that were observed in the thematic analysis, with students approaching the 2D worksheet with speed and confidence, which aligns with the low mental demand, low frustration, and high perceived success of the survey. On the other hand, the higher mental demand and frustration and lower perceived success during the AR activity aligns with the more careful, deliberate, and time-consuming approach that was demonstrated in that activity.

To contextualize these findings within the broader body of emerging technology in education literature, it is helpful to look to other fields. For example, the medical field has been a pioneer in the use of emerging visualization tools for educational training. In one study, students used virtual reality (VR) for practicing medical procedures and reported that they felt safe making mistakes and learning through trial and error [52]. Although this study used a slightly different medium (VR instead of AR), there was still an opportunity for students to independently utilize an interactive visualization tool as a learning output. This nursing study supports the claims made in this paper that it is beneficial for students to be able to make and correct their own mistakes in a low-risk environment. Another study in the astronomy field also used VR as an interactive classroom tool to teach about moon phases, and students reported that being able to "manipulate the environment" led to better ability to visualize the environment and ability to learn information. However, the study also compared alternative approaches to the learning task, including a non-digital activity and a desktop activity, and concluded that students reported those environments to be easier, faster, or less overwhelming [53]. These results support the findings of this study that immersive visualization environments may not be the easiest or quickest method of teaching, but seem to have promising potential for meaningful learning gains.

Within construction education specifically, while no exactly parallel study exists in this domain, there are some studies that can offer insight and context to the findings in this paper. A study by [18] explored student performance in a wood framing assembly task, although this task was presented physically to the students using real wood components, and what varied was their input, which consisted of either paper drawings, full-scale overlaid AR, or scaled-down AR. While this is a study of educational inputs rather than outputs, the findings offer interesting insight. They found that students who used paper resources performed the activity more quickly than those who used the scaled-down AR visualization. However, they found no difference between the paper resources and the full-scale overlay of the AR model. These findings differ from those found in this paper, indicating that there may be significant differences between using AR as an input vs. an output in construction assembly tasks. They also considered student errors, and found a wide variety in the type and frequency of errors using the three methods, but formed no conclusive determination regarding which method better supports mistake avoidance. Many students in this study mentioned that the full-scale AR assisted in selecting the correct pieces, but also mentioned that having 3D graphics overlay physical materials was distracting [18]. The work presented in this study differs from the Bloomquist study in that there was no overlay of dimensional pieces on physical pieces since the AR activity was the learning output, so although students faced challenges, distractions from the overlay of physical on virtual materials was not one of them.

In another study, students used both 2D and AR as outputs for a learning activity in an architectural design exercise focused on sustainability [40]. While not a construction assembly task, this work still allowed students to interactively create within these two media, and the researchers found that students felt they had insufficient time to complete the task, but they more frequently held this complaint when performing the task in 2D rather than AR. These findings differ from what was observed in these activities, suggesting that the discipline of the activity, or at least the requirements for student output, will have a large impact on task performance. To clarify, the architectural design study required students to create an open-ended design output, and the study from this paper required students to sequence a set of already existing pieces, so it is understandable that there would be significant differences in the performance of students. In addition to time considerations, while this activity was an open-ended design, and thus did not have an opportunity for students to make what would be considered errors or mistakes, the results did show that students were able to objectively improve their design as they iterated and created new versions within the simulated experience. These findings support the claim that AR can facilitate self-remediation where students independently alter their work to make it better.

Finally, in addition to comparisons with research on emerging visualization tools, it is helpful to draw comparisons to learning outcomes from hands-on construction building activities. For example, in a report regarding student experiences in a national design and build challenge, researchers reported that students showed technical skill development as well as an increased ability to take initiative [12]. These learning gains align with those reported in this study, where students were able to build more effectively through mistake recognition and to independently improve their performance when performing a simulated hands-on activity in AR. In fact, autonomous decision making has been reported as a key outcome of hands-on project-based learning activities in the construction domain [54], which is supported by the findings in this paper. Overall, the student performance on the AR activity presented in this work shows promising potential to replicate some of the learning gains that occur in physical hands-on activities.

## 6. Conclusions

From analysis of each of the two interventions explored, the emergent themes illustrated aspects of student performance that differed using each visualization mode as an output for documenting their work. The students who documented their construction sequences using the 2D worksheet completed the activity very quickly, but did not gener-

ally pay attention to dimensional details or make critical checks for errors. On the other hand, the students who documented their sequences using AR struggled much more while completing the activity, but demonstrated more propensity for critical checks, mistake recognition, and self-reflection during their process. Therefore, even though AR was slower and deemed to be more challenging by the students, the results suggest that AR has high potential to replicate some of the behaviors that make hands-on learning beneficial to students.

While the findings of this research may be logical, based on the differences in the experiences provided to students, the approach taken by the authors to use AR as an output for student thought remains uncommon. Instead, most prior works in this domain focus on the use of AR as an input to support students' comprehension of a design concept. The learning format chosen for AR also influenced the decision to use head-mounted (hands-free), full-scale, and fully interactive AR, characteristics which set this work apart from previous work in this domain. When compared to studies in other domains, this work supports key findings that AR has promising potential as a high-impact teaching tool, especially in providing hands-on experiences to students, but can be challenging or time-consuming. The approach used in this work illustrates ways in which documenting a construction sequence in an AR output can help students to recognize and fix their own mistakes in ways that may not be realistic to expect through 2D outputs. This may be an especially relevant skillset to develop through an AR, or otherwise virtual, mode of communication as long as comprehension of 2D documentation continues to be expected of students when they enter the construction industry, and also while the high cost of physical materials would prohibit most students from being able to physically construct a wall like the one incorporated in this research. As a result, the contribution of this work is in presenting findings that illustrate the opportunities, and potential challenges, related to using AR as an output to challenge students to produce valid construction sequences based on common 2D modes of design communication.

**Author Contributions:** Conceptualization, K.H.M., S.K.A., L.A.P., J.S.L. and W.W.; methodology, K.H.M., S.K.A., L.A.P., K.R.P., J.S.L. and W.W.; software, K.H.M. and K.R.P.; validation, K.H.M., K.R.P. and L.A.P.; formal analysis, K.H.M., L.A.P., K.R.P. and V.K.; investigation, K.H.M. and S.K.A.; resources, S.K.A.; data curation, K.H.M., L.A.P., K.R.P. and V.K.; writing—original draft preparation, K.H.M., L.A.P., K.R.P. and V.K.; writing—review and editing, K.H.M., S.K.A., L.A.P., K.R.P., V.K., J.S.L. and W.W.; visualization, K.H.M.; supervision, S.K.A.; project administration, S.K.A., J.S.L. and W.W.; funding acquisition, S.K.A., J.S.L. and W.W. All authors have read and agreed to the published version of the manuscript.

**Funding:** This material is based upon work supported by the National Science Foundation under Grant Nos. IIS-1735878 and IIS-1735804.

**Institutional Review Board Statement:** The study was conducted in accordance with the Declaration of Helsinki, and approved by the Institutional Review Board of Arizona State University (protocol code STUDY00006302, approved 21 May 2017–31 January 2023).

**Informed Consent Statement:** Informed consent was obtained from all subjects involved in the study.

**Data Availability Statement:** Some or all data, models, or code generated or used during the study are confidential in nature and may only be provided with restrictions. Data available upon request includes aggregated and anonymized survey data.

**Acknowledgments:** Thanks to Jean Salamé for work that was instrumental in the development of the AR activity and to Saadhikha Shree for assistance with video coding.

**Conflicts of Interest:** The authors declare no conflict of interest. The funders had no role in the design of the study; in the collection, analyses, or interpretation of data; in the writing of the manuscript, or in the decision to publish the results.

## Appendix A

Full set of drawings and activity provided to the students in PDF format, including (Figure A1) window elevation (Figure A2) top-of-sill floor plan (Figure A3) header floor plan and (Figure A4) windowsill floor plan.

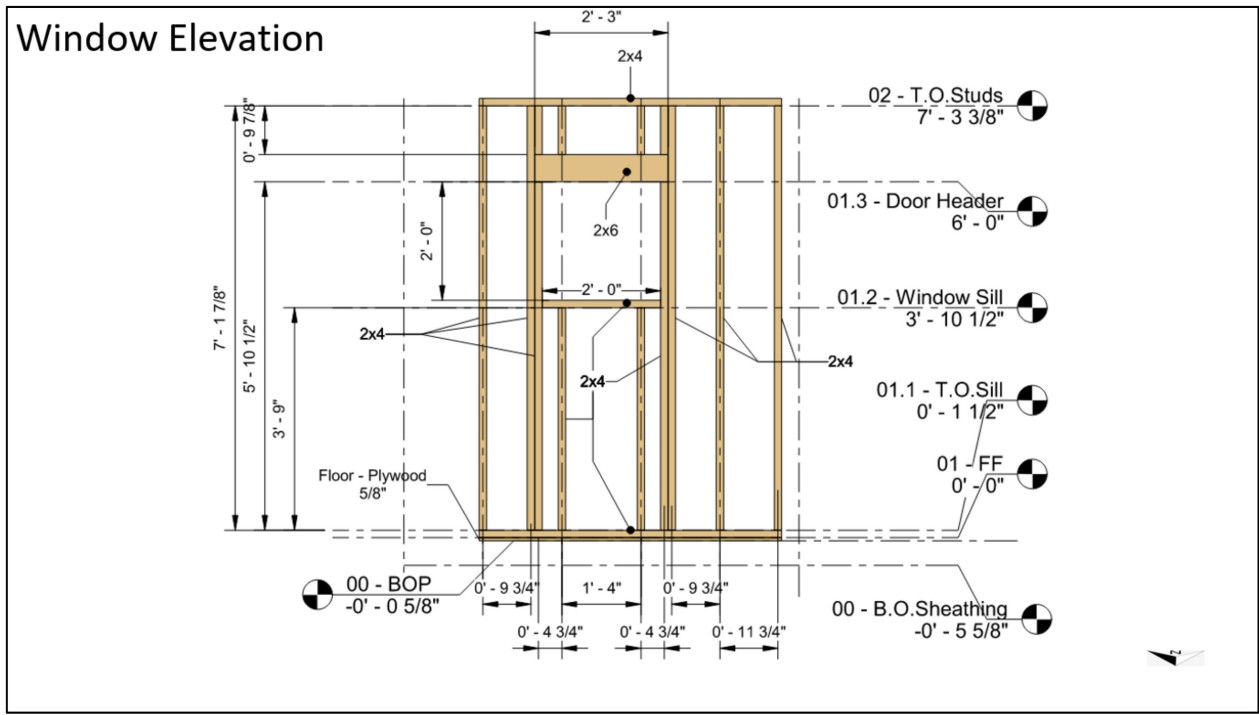

**Figure A1.** Window elevation.

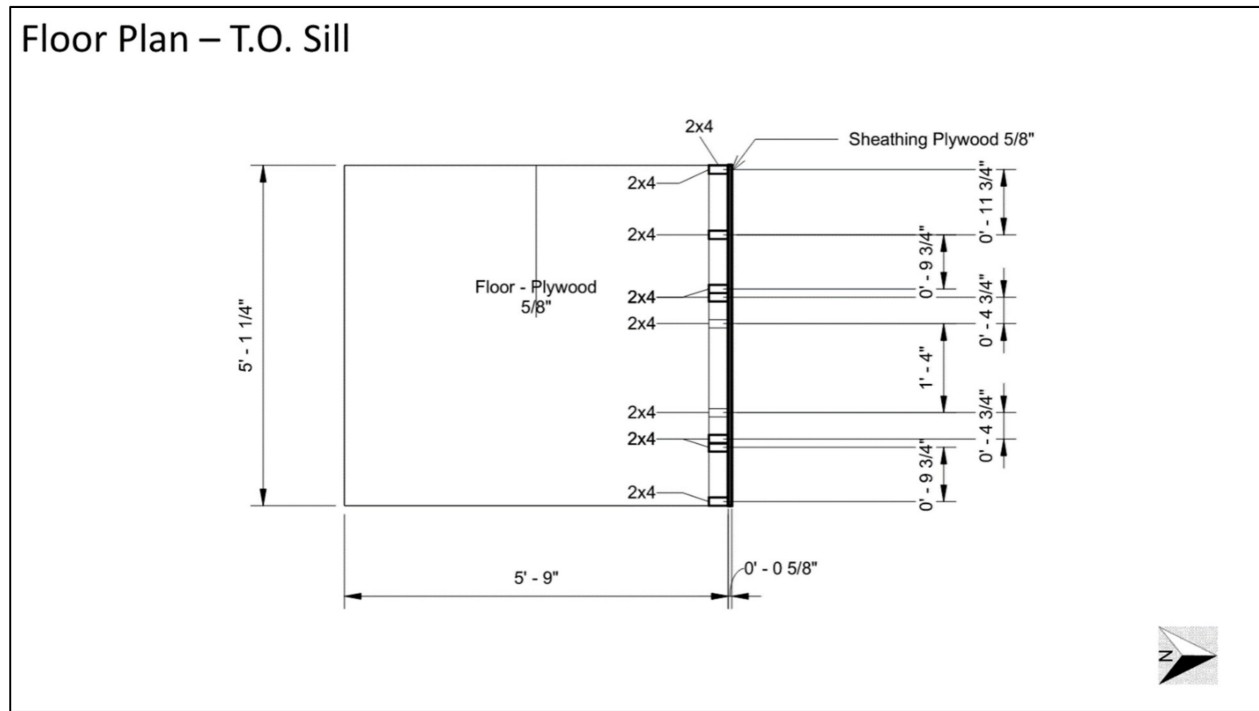

**Figure A2.** Top-of-sill floor plan.

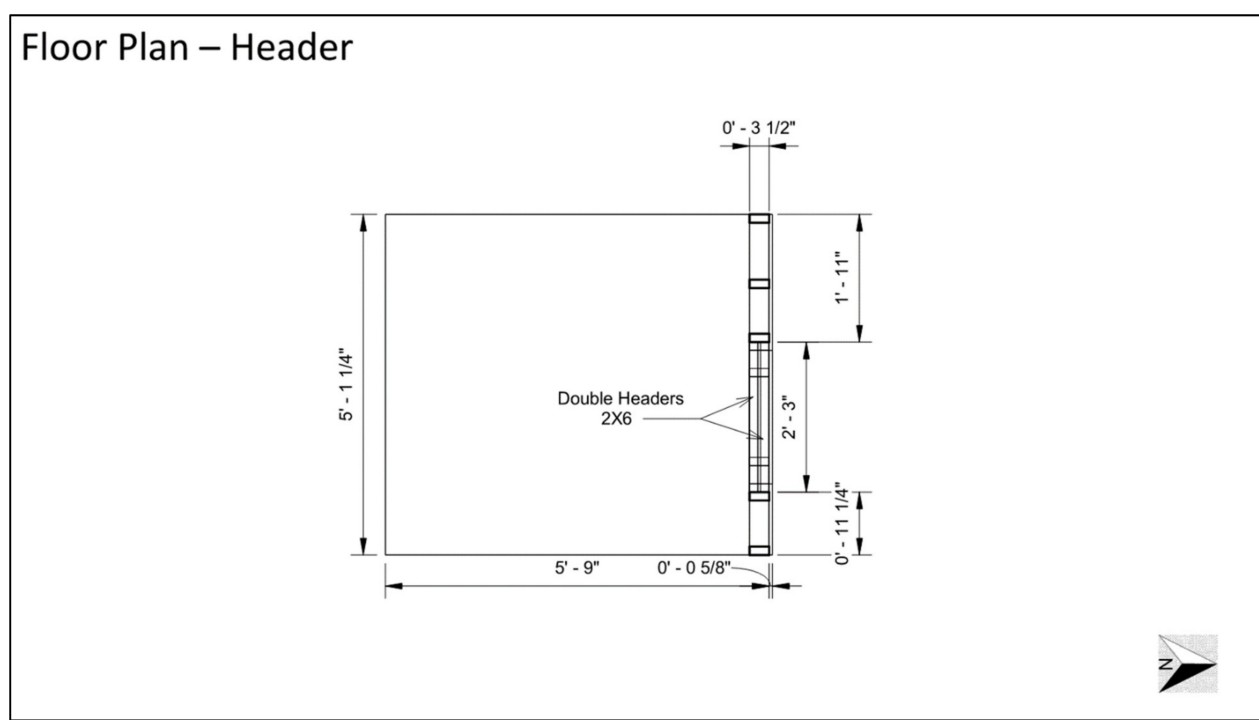

**Figure A3.** Header floor plan.

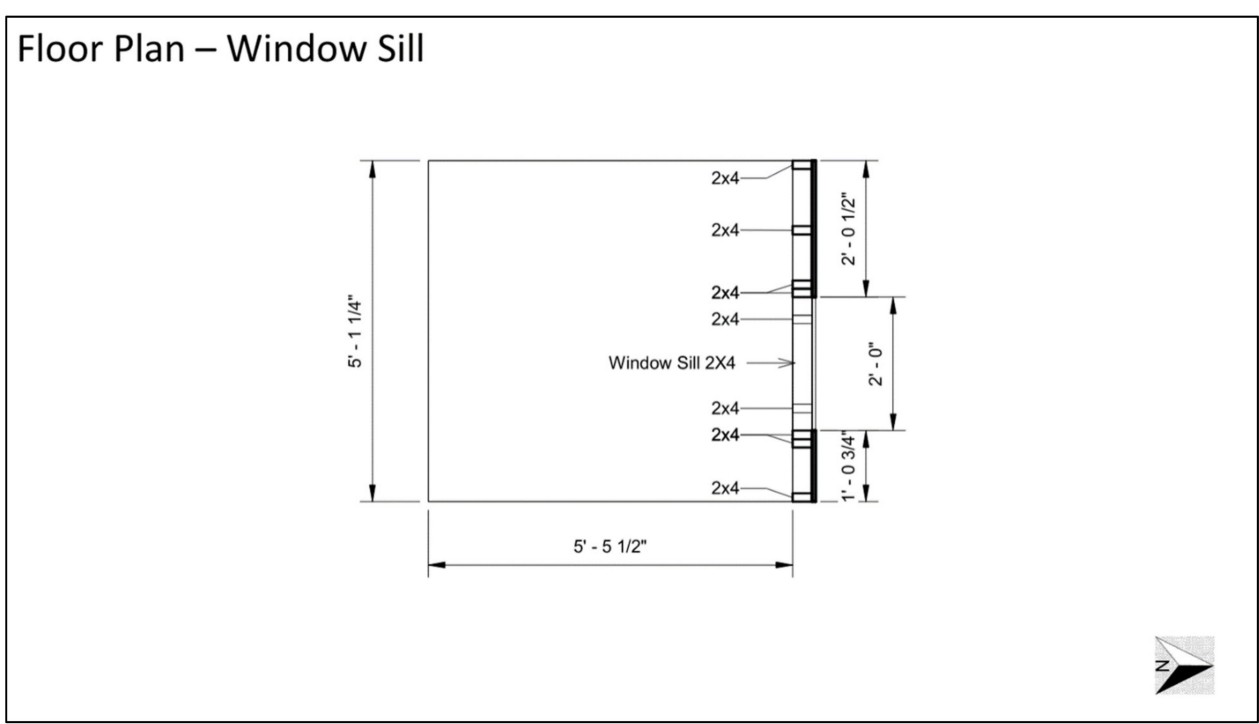

**Figure A4.** Windowsill floor plan.

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
