# Peer review of "Student Approaches and Performance in Element Sequencing Tasks Using 2D and Augmented Reality Formats"

_education, doi:10.3390/educsci12040247_

Round 1

Reviewer 1 Report

Dear Authors,
Thank you for producing this article for a possible publication. I appreciate the efforts put into describing the research methods used and the findings of this study - they are well described.
In terms of improving the quality of your work, I would highly recommend presenting a comprehensive literature review for your study. I suggest you gather more literature on the effects of visualisation in engineering education and the solutions found in previous studies. It would help you link the research gaps identified with the aim of your research.
In addition, I would strongly recommend presenting a justification for the research methods used in this study (provide a summary of similar studies conducted by previous scholars).
I would like to see more current references in your article.

I wish you all the best!
Kind regards,
Don

Reviewer 2 Report

The present manuscript reports the results of an empirical investigation addressing the performance in element sequencing tasks in civil and construction engineering research. Generally, the manuscript is in a good and mature condition, and it has the potential for an international research article reporting new findings. There are however a couple of points I would like to mention for a revised version of the manuscript:

  • It is a good approach to combine the empirical part with (three) research questions (p. 2). However, I do not see a solid chain of argumentation leading to these research questions. What are the connections between the state-of-the-art literature (derived research gaps) and the empirical study conducted for this paper? Would it make sense to change the structure so that you would use study hypotheses below the background section instead of giving research questions before presenting the background?
  • What I miss is a bit more background referring to Augmented Reality (AR). AR technologies are applied in different disciplines, and AR can be created using different devices with different (visualization) potentials, such as Head-Mounted Displays (see for e.g. https://doi.org/10.1007/s42489-021-00091-2) or mobile devices (see for e.g. https://doi.org/10.1080/10447318.2012.722466). Could you summarized the potentials of different AR techniques for educational applications, and which potentials of which AR techniques do you see for educational applications in engineering research?
  • You mix the presentation of results with a discussion of the results (section 4), and you hardly refer your study results to related studies. Could you better discriminate between the presentations of results, and your relevance of these results within the international research debate (discussion section)?
  • It would make sense to point out in your conclusion section in how far your study results back up or even contradict key research findings. You should refer to one or two key studies that are extended by your empirical study.

Round 2

Reviewer 2 Report

The author(s) provided a revised version of the manuscript and a corresponding response letter. The points mentioned in the first round are thoroughly revised. The argumentation in the response letter is sound. Against this background, I would like to recommend this version of the manuscript for publication.